# Type V Tibial Tubercle Avulsion Fracture with Suspected Complication of Anterior Cruciate Ligament Injury: A Case Report

**DOI:** 10.3390/medicina59061061

**Published:** 2023-06-01

**Authors:** Hiroki Okamura, Hiroki Ishikawa, Takuya Ohno, Shogo Fujita, Kei Nagasaki, Katsunori Inagaki, Yoshifumi Kudo

**Affiliations:** 1Department of Orthopedic Surgery, Nihon Koukan Hospital, 1-2-1 Koukandori, Kawasaki 210-0852, Japan; hiroki.f.marinos@gmail.com (H.I.); shinenow44@yahoo.jp (T.O.); fala500104@yahoo.co.jp (S.F.); keppoko@hotmail.com (K.N.); 2Department of Orthopedic Surgery, Showa University School of Medicine, 1-5-8 Hatanodai, Shinagawa-ku, Tokyo 142-8666, Japan; katsu@med.showa-u.ac.jp (K.I.); kudo_4423@yahoo.co.jp (Y.K.)

**Keywords:** tibial tubercle avulsion fractures, anterior cruciate ligament, knee, sports injury

## Abstract

*Background and Objectives*: Type V tibial tubercle avulsion fractures are extremely rare; therefore, information on them remains limited. Furthermore, although these fractures are intra-articular, to the best of our knowledge, there are no reports on their assessment via magnetic resonance imaging (MRI) or arthroscopy. Accordingly, this is the first report to describe the case of a patient undergoing detailed evaluation via MRI and arthroscopy. *Case Presentation*: A 13-year-old male adolescent athlete jumped while playing basketball, experienced discomfort and pain at the front of his knee, and fell down. He was transported to the emergency room by ambulance after he was unable to walk. The radiographic examination revealed a Type Ⅴ tibial tubercle avulsion fracture that was displaced. In addition, an MRI scan revealed a fracture line extending to the attachment of the anterior cruciate ligament (ACL); moreover, high MRI intensity and swelling due to ACL were observed, suggesting an ACL injury. On day 4 of the injury, open reduction and internal fixation were performed. Furthermore, 4 months after surgery, bone fusion was confirmed, and metal removal was performed. Simultaneously, an MRI scan obtained at the time of injury revealed findings suggestive of ACL injury; therefore, an arthroscopy was performed. Notably, no parenchymal ACL injury was observed, and the meniscus was intact. The patient returned to sports 6 months postoperatively. *Conclusion*: Type V tibial tubercle avulsion fractures are known to be extremely rare. Based on our report, we suggest that MRI should be performed without hesitation if intra-articular injury is suspected.

## 1. Introduction

Tibial tubercle avulsion fractures are known to be relatively rare in adolescents, accounting for 0.4–2.7% of all pediatric fractures and <1% of all epiphysis injuries [1,2].

In 1955, Watson–Jones first reported the fracture morphology of tibial tubercle avulsion fractures and classified them from Type I to Type III. In 1980, Ogden et al. refined the classification to include subtypes according to the degree of displacement and comminution, and this classification has been widely used to date. Subsequently, in 1985, Ryu and Debenhum added Type IV, and in 2003, Mackoy added Type V to the classification [3,4]. According to a previous study, conservative treatment should be considered for Types Ⅰ and Ⅱ, and surgery is preferable for Type Ⅲ and above if a dislocation is present [5]. In particular, Type V fracture involves the posterior bone cortex as well as the articular surface, and this fracture requires accurate repositioning, firm fixation, and careful postoperative therapy. Moreover, Type V fracture morphology is very rare and combines the morphology of Ogden Type IIIB with Salter–Haris Type IV [4]. Notably, there are very few reports of Ogden Type V fractures, with only five cases reported to date; hence, information on these fractures is limited [4,6,7]. Furthermore, although these fractures are intra-articular, to the best of our knowledge, there have been no reports of them being assessed via magnetic resonance imaging (MRI) or arthroscopy.

Here, we report our experience with a very rare case of Ogden Type V fracture, along with a literature review.

## 2. Case Report

A 13-year-old male adolescent athlete presented to our hospital’s emergency room with right knee pain. He reported that he jumped while playing basketball, felt discomfort and pain at the front of his knee, and fell down. Furthermore, he had become unable to walk and was transported to the emergency room by ambulance. Notably, he presented with no past medical history.

An orthopedic physical examination of the right knee revealed tenderness from the patella tendon to the tibial tuberosity. Moreover, swelling was noted around the knee and proximal leg. In addition, knee joint effusion was noted. Notably, active flexion of the right knee was not possible, and the knee had a limited range of motion (ROM). Furthermore, the ligament stability test could not be performed because of pain. Nonetheless, the neurovascular status was intact.

Radiographic examination revealed a displaced Type Ⅴ tibial tubercle avulsion fracture (Figure 1). Furthermore, the patient was hospitalized with splint immobilization, and his condition was carefully monitored until surgery. CT scan revealed a Type Ⅴ tibial tubercle avulsion fracture (Figure 2a). In addition, an MRI scan revealed that the fracture line reached the attachment of the ACL, and high MRI intensity and swelling due to the ACL were observed, suggesting an ACL injury (Figure 2b). No meniscus or other ligament damage was noted. Fortunately, compartment syndrome was not developed, and open reduction and internal fixation were performed on day 4 of the injury. However, periosteum disruption and patella tendon rupture were observed (Figure 3). The fracture was fixed with a cannulated cancellous screw (CCS), and the disrupted periosteum and patella tendon were repaired (Figure 4).

The patient was non-weight bearing, 1/3-weight bearing, 1/2-weight bearing, 2/3-weight bearing, and full-weight bearing 6, 6, 8, 10, and 12 weeks after surgery, respectively. Furthermore, postoperatively, after 3 weeks of splint immobilization, the patient started ROM exercises. Furthermore, full ROM was achieved 8 weeks after surgery, and the patient was able to sit upright. Anterior drawer and Lachman tests were negative. Four months after surgery, bone fusion was confirmed, and the patient underwent metal removal (Figure 5). Meanwhile, an MRI scan obtained at the time of injury revealed findings that indicated ACL injury, and based on the wishes of the patient’s family, arthroscopy was performed after obtaining consent from the patient (Figure 6). Notably, no parenchymal ACL injury was observed, and the meniscus was intact. The patient returned to sports at 6 months postoperatively. His knee injury and osteoarthritis outcome score at 6 months postoperatively was 100, and there was no evidence of premature closure of the proximal tibial epiphysis or growth retardation at this time.

## 3. Discussion

In adolescents, tibial tubercle avulsion fractures are relatively rare and account for 0.4–2.7% of all pediatric fractures and <1% of all epiphysis injuries [1,2]. At present, the fracture morphology of tibial tubercle avulsion fractures has been widely reported and used to classify fractures into Types I–V [8]. Notably, the morphology of Ogden Type V fracture is characterized by the morphology of Ogden Type IIIB fracture combined with that of Salter–Haris Type IV fracture, and this is a very rare fracture morphology [4]. However, the information on Ogden Type V fracture remains limited because of the scarcity of reports on this fracture, with only five cases reported to date [4,6,7]. Furthermore, despite the fact that this is an intra-articular fracture, there have been no reports of its evaluation using MRI or arthroscopy. To the best of our knowledge, this is the first detailed case report of a patient with Ogden Type V fracture evaluated via MRI and arthroscopic findings.

The ossification of tibial tuberosity can be classified into four stages: the cartilaginous, apophyseal, epiphyseal, and osseous stages [9]. In particular, during the epiphyseal stage, traction-sensitive physeal hypertrophic columnar cartilage replaces fibrocartilage [9]. This osseous pattern and apophyseal stage can create a mechanically weak area, facilitating the avulsion of the tibial tuberosity [4,10]. The epiphyseal stage is typically observed in girls aged 10–15 years and boys aged 11–17 years. Notably, tubercle avulsion fracture usually occurs during 8–15 years of age, and men are 10 times more likely to be injured than women [11,12,13]. In the present case, tibial tuberosity injury was noted at the epiphyseal stage when the patient was aged 13 years, and the tuberosity was vulnerable to traction forces.

X-ray imaging has been used for diagnostic evaluation and classification of fractures. However, Pace et al. reported that further evaluation with CT or MRI is necessary owing to the possibility of complications such as quadriceps rupture, patella tendon rupture, and intra-articular tissue damage [7]. Moreover, MRI is recommended for similar intra-articular fractures, such as avulsion fracture of tibia eminence and fracture of tibia plateau, to examine the damage to the intra-articular tissues, including ACL [14,15]. Conversely, to the best of our knowledge, no studies on Ogden Type V fractures have evaluated the intra-articular region using MRI or arthroscopy to date. In our patient, an MRI scan revealed that the fracture line had reached the ACL attachment area, indicating the possibility of an ACL injury. Arthroscopy findings revealed no damage to the intra-articular ligaments or meniscus at the time of metal removal. In the future, MRI should be performed for patients exhibiting fracture lines on the articular surface.

It has been reported that treatment for Type Ⅰ and Ⅱ fractures should be conservative; conversely, surgery is preferred for Type Ⅲ and above fractures if a dislocation is present [5]. Notably, CCS fixation is the most common surgical fixation method, although some studies have reported that plate fixation may be necessary [4,7]. However, in one of these studies, plate fixation was also found to be associated with early closure of the proximal tibial epiphysis [7]. In the present case, CCS provided good fixation force and outcomes. Furthermore, it has been reported that postoperative treatment should include the following: 4 weeks of immobilization, 2 weeks of knee brace application, and initiation of extensor mechanism strengthening after the removal of the knee brace. However, only a few standardized protocols are available for postoperative treatment [16]. In particular, Type V fracture involves both the posterior bone cortex and articular surface, and it requires cautious loading. In the present case, we applied a knee brace after 3 weeks of immobilization, started partial weight bearing at 6 weeks postoperatively, and achieved full-weight bearing and full flexion at 12 weeks postoperatively. Furthermore, after confirming that the muscles had regained their strength, the patient was allowed to return to sports in approximately 6 months. The present study reports novel findings with a detailed examination and treatment outcome of a very rare case of Ogden Type V fracture. To the best of our knowledge, no studies have reported on MRI-based evaluation of Ogden Type V fractures. In the present case, we found that the fracture line to the articular surface had reached the attachment of the ACL on an MRI scan; moreover, high MRI intensity and swelling due to ACL were observed, potentially causing parenchymal ACL injury. Finally, an arthroscopic evaluation was performed to assess intra-articular damage, including parenchymal damage to ACL, and this enabled our patient to successfully return to sports.

Previous studies have reported that perioperative complications of tibial tubercle fractures include bursitis and tenderness, refractures, wound infection, leg length differences associated with early closure of the proximal tibial epiphysis, and compartment syndrome [2,7,17,18,19,20]. Of these, the most common complication is bursitis, accounting for 56% of all complications, and it requires metal removal [18,20]. Notably, the complications specific to the adolescent age group include early closure of the proximal tibial epiphysis and the associated limb length difference and angular deformity [18]. According to Gautier et al., these complications are caused by growth acceleration or retardation at the proximal tibial epiphysis [18]. Moreover, Pace et al. reported early closure of the proximal tibial epiphysis in a patient for whom plate fixation was used [7]. In particular, compartment syndrome is a serious complication [2,8,19] caused by the disruption of branches of the anterior recurrent tibial artery, and it is reported to occur in 20% of cases of tibial tubercle fractures, especially in adolescents [8,19]. Burkhart et al. reported a case of a patient with compartment syndrome requiring below-knee amputation, a complication of particular concern [17]. In the present case, fortunately, none of these complications occurred at this stage.

## 4. Conclusions

Type V tibial tubercle avulsion fractures are extremely rare. To the best of our knowledge, this is the first report to describe a case of tubercle avulsion fractures with detailed evaluation using MRI and arthroscopy. Based on our report, if intra-articular injury is suspected, an MRI evaluation should be performed without hesitation.

## 5. Clinical Message

MRI should be performed to evaluate the presence of concomitant injuries, such as periprosthetic ligament and intra-articular injuries.

## Figures and Tables

**Figure 1 medicina-59-01061-f001:**
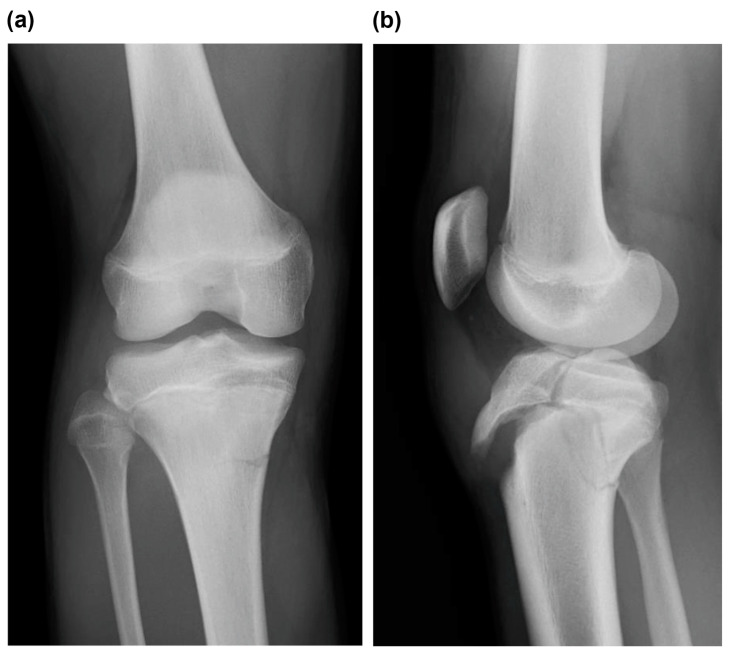
(**a**) Frontal view and (**b**) lateral view of plain radiographs obtained at the time of injury indicated Type Ⅴ tibial tubercle avulsion fracture.

**Figure 2 medicina-59-01061-f002:**
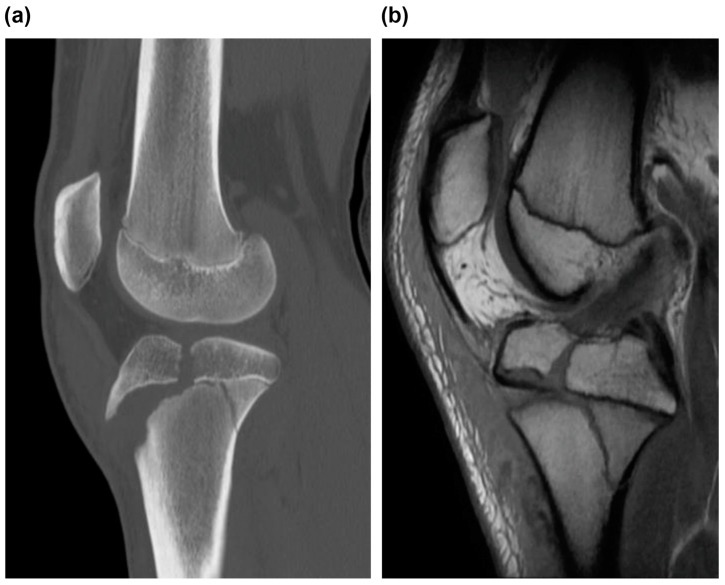
(**a**) CT and (**b**) MRI scans of the right knee, sagittal view. (**a**) CT scan revealed tibial tubercle fracture of Ogden Type IIIB and Salter–Harris Type IV, indicating tibial tubercle fracture of Ogden Type V. (**b**) MRI scan revealed that the fracture line reached the attachment of ACL, and high MRI intensity and swelling due to ACL were observed, suggesting an ACL injury. CT, computed tomography; MRI, magnetic resonance image; and ACL, anterior cruciate ligament.

**Figure 3 medicina-59-01061-f003:**
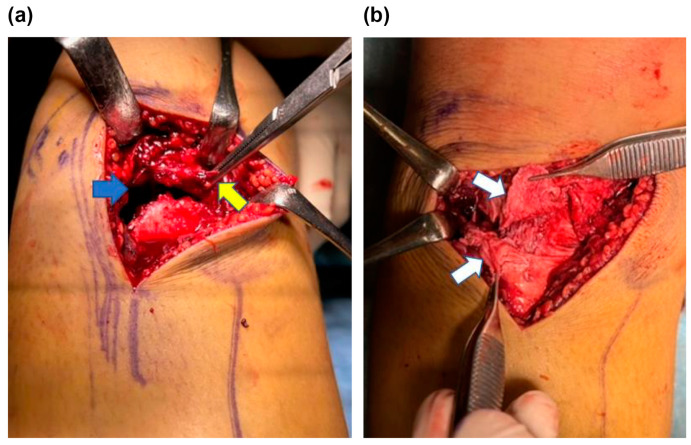
Intraoperative image of the right knee fracture area. (**a**) Disruption of the periosteum (yellow arrow) embedded in the fracture site (blue arrow). (**b**) Partial rupture of the patellar tendon (white arrow) was noted.

**Figure 4 medicina-59-01061-f004:**
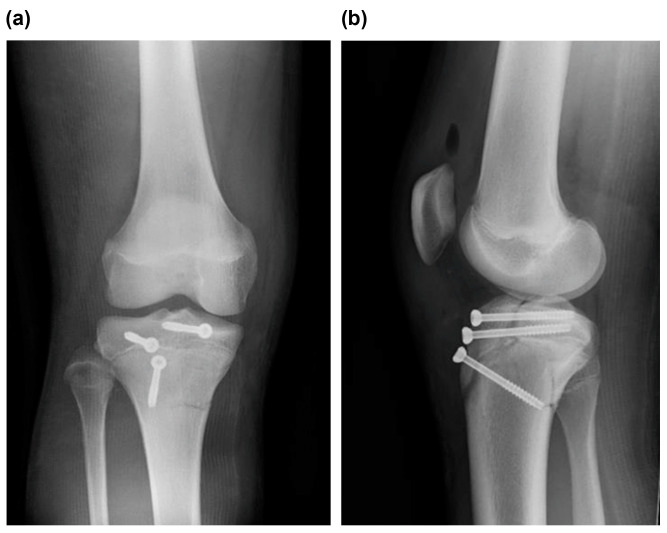
(**a**) Frontal view and (**b**) lateral view of postoperative plain radiographs revealed good fixation by repositioning.

**Figure 5 medicina-59-01061-f005:**
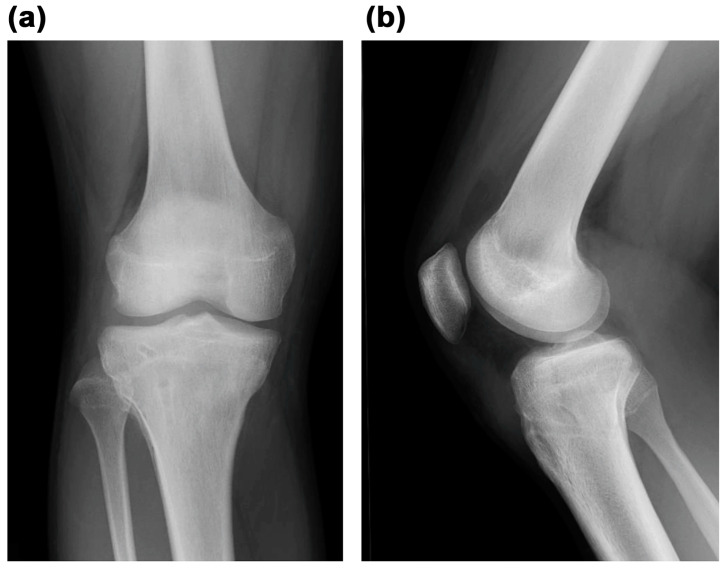
(**a**) Frontal view and (**b**) lateral view of plain radiographs obtained after metal removal revealed good bone union.

**Figure 6 medicina-59-01061-f006:**
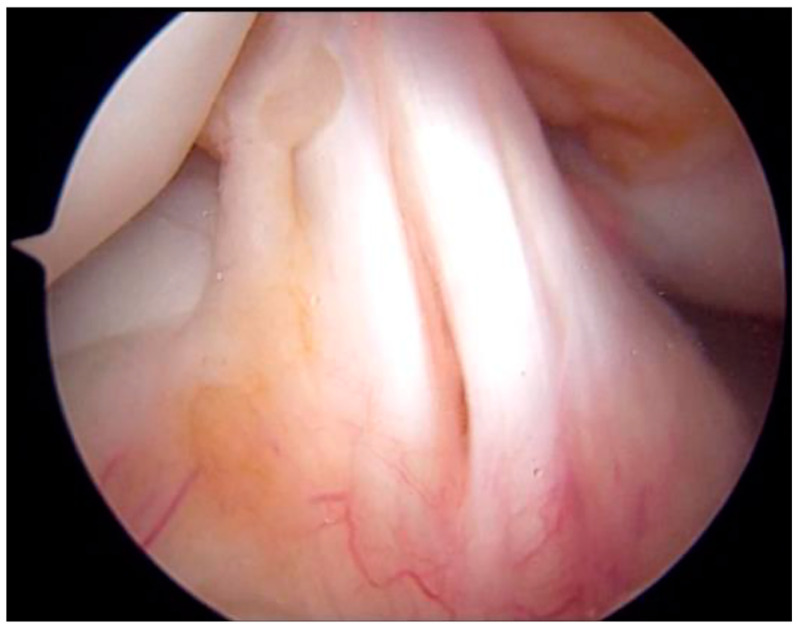
Arthroscopic image of the ACL in the right knee. ACL was segmented with no synovial coverage, but there was no obvious tearing. ACL, anterior cruciate ligament.

## Data Availability

The data presented in this study are available on request from the corresponding author. The data are not publicly available due to them containing information that could compromise the privacy of research participants.

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
