# Peer review of "Type V Tibial Tubercle Avulsion Fracture with Suspected Complication of Anterior Cruciate Ligament Injury: A Case Report"

_medicina, 2023, doi:10.3390/medicina59061061_

Round 1

Reviewer 1 Report

Dear Authors,

I was pleased to review the paper entitled "Type V Tibial Tubercle Avulsion Fracture with Suspected 2 Complication of Anterior Cruciate Ligament Injury: A Case 3 Report”

This article aims to enhance the importance of the evaluation of these kind of fractures by MRI in order to assess related soft tissue damage and choose the proper treatment for the patient.

I read the article with interest, the topic of the study is interesting and useful.

According to my opinion, some minor changes are needed to be considered suitable for publication:

-The abstract is adequately developed and is useful to frame the characteristics and purpose of the study. 

Line 20, don't start a sentence with an acronym.

Line 26, explicate acronyms if used for the first time “…to sports at 6M postoperatively…”

-The introduction is complete and comprehensive. 

Briefly introduce the types of treatment according to the classification.

- Case report. Lines 79-80, Why did you decide to perform an arthroscopy? did you perform a thorough clinical evaluation that revealed a positive anterior drawer or lachman test?

Include that informed consent was obtained to perform the procedures

-The discussion is sufficiently developed and the result well written. 

The role of imaging is critical in knee pathology (from Medicina-MDPI DOI: 10.3390/medicina58091164).

Line 129, don't start a sentence with an acronym “CCS fixation”.

- The conclusion is well elaborated.

Review the English language throughout the text.

to correct minor errors.

Author Response

Dear Authors,

I was pleased to review the paper entitled "Type V Tibial Tubercle Avulsion Fracture with Suspected 2 Complication of Anterior Cruciate Ligament Injury: A Case 3 Report”

This article aims to enhance the importance of the evaluation of these kind of fractures by MRI in order to assess related soft tissue damage and choose the proper treatment for the patient.

I read the article with interest, the topic of the study is interesting and useful.

According to my opinion, some minor changes are needed to be considered suitable for publication:

-The abstract is adequately developed and is useful to frame the characteristics and purpose of the study.

Line 20, don't start a sentence with an acronym.

Line 26, explicate acronyms if used for the first time “…to sports at 6M postoperatively…”

-The introduction is complete and comprehensive.

Briefly introduce the types of treatment according to the classification.

- Case report. Lines 79-80, Why did you decide to perform an arthroscopy? did you perform a thorough clinical evaluation that revealed a positive anterior drawer or lachman test?

Include that informed consent was obtained to perform the procedures

-The discussion is sufficiently developed and the result well written.

The role of imaging is critical in knee pathology (from Medicina-MDPI DOI: 10.3390/medicina58091164).

Line 129, don't start a sentence with an acronym “CCS fixation”.

- The conclusion is well elaborated.

Review the English language throughout the text.

Response to Reviewer1

Thank you for your insightful comments. We appreciate your time and suggestions on our manuscript. Our responses to the comments from Reviewer 1 are as follows:

Comment (1)

Line 20, don't start a sentence with an acronym.

Response (1)

Thank you for pointing this out. We have revised the following text as per your comment:

In addition, MRI scan revealed a fracture line extending till the attachment of the anterior cruciate ligament (ACL); moreover, high MRI intensity and swelling due to ACL were observed, suggesting an ACL injury. (P.1, Line21-23 )

Comment (2)

Line 26, explicate acronyms if used for the first time “…to sports at 6M postoperativel.

Response (2)

Thank you for pointing this out. We have revised the following text as per your comment:

The patient returned to sports at 6 months postoperatively. (P1., Line27-28)

Comment (3)

Briefly introduce the types of treatment according to the classification.

Response (3)

We agree that treatment according to the classification should be added; accordingly, we have added the following text:

It has been reported that treatment should be conservative for TypeⅠ and Ⅱ, and surgery is preferable for Type Ⅲ and above if dislocation is present[13]. In particular, According to a previous study, conservative treatment should be considered for Types Ⅰ and Ⅱ, and surgery is preferable for Type Ⅲ and above if dislocation is present [5]. (P.6, Line148-150)

Comment (4)

Case report. Lines 79-80, Why did you decide to perform an arthroscopy? did you perform a thorough clinical evaluation that revealed a positive anterior drawer or lachman test?

Include that informed consent was obtained to perform the procedures

Response (4)

Thank you for your valuable comment.

At the time of injury, some MRI findings were suggestive of ACL injury, and the patient requested to undergo arthroscopy. No other findings suggested ACL injury.

Based on your comment, we revised the sentence as follows:

Anterior drawer and Lachman tests were negative. (P.4, Line98)

Meanwhile, an MRI scan obtained at the time of injury revealed findings that indicated ACL injury, and based on the wish of the patient’s family, arthroscopy was performed after obtaining consent from the patient (P.4, Line100-102)

Comment (5)

Line 129, don't start a sentence with an acronym “CCS fixation”.

Response (5)

Thank you for pointing this out. We have made the following revisions based on your comment:

Notably, CCS fixation is the most common surgical fixation method, although some studies have reported that plate fixation may be necessary (P.6, Line150-151)

Comment (5)

Review the English language throughout the text.

Response (5)

Thank you for your valuable comment. We have carefully rechecked the entire manuscript for English; moreover, the manuscript has been checked by a native English speaker.

Reviewer 2 Report

Dear authors,

I had the pleasure to read your manuscript. 

Looking at figure 2, I think the fracture description requires more details. There is also a fracture line descending posteriorly. Was this also described in the previous case reports ?

Line 59: obscured parenchyma, is not very medical in my eyes. Do you mean hyperintense signal / hematoma/ swelling of the ACL ? was there discontinuation ? any indirect sign of an ACL tear ?( bone bruise, kinking/buckling of the PCL). If the ACL footprint is in the fracture line, and the fragment is displaced, it can be expected that the ACL becomes a "wavy" appearance (similar to an avulsion fracture of the tibial eminence). Where is the surprise ? 

Line 79: you mention a "possibility" of an ACL rupture. What do you mean by "possibility" ? After any trauma there is a possibility of injury. Was there a clinical suspicion on clinical examination ? or simply the MRI at the time of injury ? (see comment above, that it is not a surprise to see a wavy pattern of the ACL in this case, nor hyperintense signal). 

Line 115: you mention here that in your case the tibial tuberosity was injured  at the age of 13 years, but in your case presentation you mention the age of 15 years. 

Concerning your conclusion: MRI had no impact on the patient's outcome, nor on your treatment. 

Author Response

Dear authors,

I had the pleasure to read your manuscript.

Looking at figure 2, I think the fracture description requires more details. There is also a fracture line descending posteriorly. Was this also described in the previous case reports?

Line 59: obscured parenchyma, is not very medical in my eyes. Do you mean hyperintense signal / hematoma/ swelling of the ACL? was there discontinuation? any indirect sign of an ACL tear?( bone bruise, kinking/buckling of the PCL). If the ACL footprint is in the fracture line, and the fragment is displaced, it can be expected that the ACL becomes a "wavy" appearance (similar to an avulsion fracture of the tibial eminence). Where is the surprise?

Line 79: you mention a "possibility" of an ACL rupture. What do you mean by "possibility"? After any trauma there is a possibility of injury. Was there a clinical suspicion on clinical examination? or simply the MRI at the time of injury? (see comment above, that it is not a surprise to see a wavy pattern of the ACL in this case, nor hyperintense signal).

Line 115: you mention here that in your case the tibial tuberosity was injured at the age of 13 years, but in your case presentation you mention the age of 15 years.

Concerning your conclusion: MRI had no impact on the patient's outcome, nor on your treatment.

Response to Reviewer2

Thank you for your insightful comments. We appreciate your time and suggestions on our manuscript. Our responses to the comments from Reviewer 2 are as follows:

Comment (1)

Looking at figure 2, I think the fracture description requires more details. There is also a fracture line descending posteriorly. Was this also described in the previous case reports?

Response (1)

We agree that the indicated point requires more detailed description of the fracture. Further, as pointed out by you, a fracture line was found to be descending posteriorly, which was reported as a fracture line of Salter–Harris IV. To the best of our knowledge, this is a very rare fracture, with only five cases reported since the first report in 2003 [McKoy, B.E.; Stanitski, C.L. Acute tibial tubercle avulsion fractures. Orthop Clin North Am 2003, 34, 397–403. DOI: 10.1016/s0030-5898(02)00061-5.]. Based on your comment, we have revised the indicated sentence as follows:

CT scan revealed tibial tubercle fracture of Ogden Type IIIB and Salter–Harris Type IV, indicating tibial tubercle fracture of Ogden Type V.  (P.3, Line81-83 )

Comment (2)

Line 59: obscured parenchyma, is not very medical in my eyes. Do you mean hyperintense signal / hematoma/ swelling of the ACL? was there discontinuation? any indirect sign of an ACL tear?( bone bruise, kinking/buckling of the PCL). If the ACL footprint is in the fracture line, and the fragment is displaced, it can be expected that the ACL becomes a "wavy" appearance (similar to an avulsion fracture of the tibial eminence). Where is the surprise?

Response (2)

Thank you for your valuable comment. We agree that the term was misleading. We meant high MRI intensity and swelling due to ACL. At our institution, we do not consider indirect signs, such as lateral notch, segon fracture, and kinking/buckling of the PCL. It has been recommended that MRI should be performed in cases of tuberosity avulsion fractures where the fracture line reaches to the joint owing to the fact that the intra-articular tissues such as ligaments and meniscus may be damaged [Pace, J.L.; McCulloch, P.C.; Momoh, E.O.; Nasreddine, A.Y.; Kocher, M.S. Operatively treated type IV tibial tubercle apophyseal fractures. J Pediatr Orthop 2013, 33, 791–796. DOI: 10.1097/BPO.0b013e3182968984]. However, only a few previous studies have reported on Ogden Type V fractures with intra-articular evaluation. Moreover, it has been reported that an avulsion fracture of the tibial eminence or tibia plateau fracture may be accompanied by ACL injury [Ishibashi Y, Tsuda E, Sasaki T, Toh S. Magnetic resonance imaging AIDS in detecting concomitant injuries in patients with tibial spine fractures. Clinical orthopaedics and related research. 2005;(434):207-12. Epub 2005/05/03. doi: 10.1097/00003086-200505000-00031.] [Yamauchi S, Sasaki S, Kimura Y, Yamamoto Y, Tsuda E, Ishibashi Y. Tibial eminence fracture with midsubstance anterior cruciate ligament tear in a 10-year-old boy: A case report. International journal of surgery case reports. 2020;67:13-7. Epub 2020/01/29. doi: 10.1016/j.ijscr.2019.12.036.], although the origin of injury and type of fracture may differ. Therefore, it is important to suspect ACL injury and perform a detailed MRI. We believe that our study is important because it is the first report of a detailed intra-articular evaluation of tibial tubercle Ogden type V fracture.

Based on your comment, we revised the sentence as follows:

MRI scan revealed that the fracture line reached the attachment of ACL, and high MRI intensity and swelling due to ACL were observed, suggesting an ACL injury. (P.3, Line83-84 )

Moreover, MRI is recommended for similar intra-articular fractures, such as avulsion fracture of tibia eminence and fracture of tibia plateau, to examine the damage to the intra-articular tissues, including ACL (P.5, Line139-141)

Comment (3)

Line 79: you mention a "possibility" of an ACL rupture. What do you mean by "possibility"? After any trauma there is a possibility of injury. Was there a clinical suspicion on clinical examination? or simply the MRI at the time of injury? (see comment above, that it is not a surprise to see a wavy pattern of the ACL in this case, nor hyperintense signal).

Response (3)

Thank you for your valuable comment. We agree that the term was misleading.

At the time of injury, some MRI findings were suggestive of ACL injury, and the patient requested to undergo arthroscopy. No other findings suggested ACL injury.

Based on your comment, we revised the sentence as follows:

Meanwhile, an MRI scan obtained at the time of injury revealed findings that indicated ACL injury, and based on the wish of the patient’s family, arthroscopy was performed after obtaining consent from the patient (P.4, Line100-102)

Comment (4)

Line 115: you mention here that in your case the tibial tuberosity was injured at the age of 13 years, but in your case presentation you mention the age of 15 years.

Response (4)

We apologize for the confusion and thank you for pointing this out. The patient is 13 years old. To address your comment, we have made the following changes:

A 13-year-old male adolescent athlete arrived at our hospital emergency room with right knee pain. (P.2, Line54-55)

Comment (5)

Concerning your conclusion: MRI had no impact on the patient's outcome, nor on your treatment.

Response (5)

Indeed, the results may not have affected the outcome of the patient. However, we believe that a detailed intra-articular evaluation enabled the patient to return to sports without anxiety.

Round 2

Reviewer 1 Report

The authors correctly responded to the reviews.